# Stochastic Multi-Objective Multi-Armed Bandits: Regret Definition and Algorithm

**Mansoor Davoodi**                                                                 *Mansoor.DavoodiMonfared@here.com*
[1] *Faculty of Electrical Engineering and Information Technology, Ruhr-University Bochum, Bochum 44801, Germany*
[2] *Department of Computer Science and Information Technology, Institute for Advanced Studies in Basic Sciences,*
*Prof. Yousef Sobouti Blvd. 444, 45137-66731, Zanjan, Iran*
[3] *HERE Deutschland GmbH Co. KG, Invalidenstraße 116, 10115 Berlin, Germany*
**Setareh Maghsudi**                                                       *Setareh.Maghsudi@ruhr-uni-bochum.de*
*Faculty of Electrical Engineering and Information Technology*
*Ruhr-University Bochum, Bochum 44801, Germany*

**Reviewed on OpenReview:** *https://openreview.net/forum?id=7N7sK5CFuP*

## Abstract

Multi-armed bandit (MAB) problems are widely applied to online optimization tasks that require balancing exploration and exploitation. In practical scenarios, these tasks often involve multiple conflicting objectives, giving rise to multi-objective multi-armed bandits (MO-MAB). Existing MO-MAB approaches predominantly rely on the Pareto regret metric introduced in Drugan & Nowé (2013). However, this metric has notable limitations, particularly in accounting for all Pareto-optimal arms simultaneously. To address these challenges, we propose a novel and comprehensive regret metric that ensures balanced performance across conflicting objectives. Additionally, we introduce the concept of *Efficient Pareto-Optimal* arms, which are specifically designed for online optimization. Based on our new metric, we develop a two-phase MO-MAB algorithm that achieves sublinear regret for both Pareto-optimal and efficient Pareto-optimal arms.

## 1 Introduction

Multi-armed bandits (MAB) form a foundational framework in machine learning with wide-ranging applications in online optimization tasks such as recommendation systems and online advertising Slivkins (2019). The objective in MAB problems is to sequentially select actions (arms) that balance exploration (learning about unknown arms) and exploitation (maximizing cumulative rewards). In many real-world scenarios, decisions involve trade-offs among multiple conflicting objectives, which has led to the study of multi-objective multi-armed bandits (MO-MAB). In MO-MAB, the goal is to identify Pareto-optimal (PO) solutions that represent the best trade-offs among the objectives, making this problem highly relevant in areas such as multi-criteria decision-making Wei et al. (2021).

Despite growing interest in MO-MAB, most existing work relies on the Pareto regret metric introduced by Drugan et al. Drugan & Nowé (2013). Although this metric serves as a useful starting point and has been widely adopted, it has several significant limitations. Specifically, the metric focuses on the minimum distance of an arm's reward vector to the Pareto front in one direction, neglecting the performance across other objectives. This can lead to situations where algorithms optimizing a single objective are evaluated as highly effective, even though they may fail to balance other objectives (e.g., see Xu & Klabjan (2023)). Moreover, the metric inadequately penalizes poor performance in unoptimized objectives and fails to ensure diversity in the objective space (for a detailed discussion of this metric, see Section 3). Consequently, existing regret metrics are insufficient for fully evaluating the performance of MO-MAB algorithms.

To address these limitations, a more comprehensive regret definition that accounts for multiple objectives is crucial for a more accurate evaluation of algorithms in multi-objective settings. In this paper, we make the following key contributions:

- We propose a novel and comprehensive regret metric for MO-MAB that overcomes the limitations of existing metrics by simultaneously considering all objectives.

- We introduce the concept of *Efficient Pareto-Optimal* (EPO) arms, tailored for online optimization settings, to capture the round-based nature of MO-MAB problems better.

- We develop a two-phase explore-exploit algorithm that achieves sublinear regret for both PO and EPO arms. For $n$ arms over $T$ rounds, it offers two implementations: an exponential-time variant with regret $O\left(T^{\frac{2}{3}}(n\log T)^{\frac{1}{3}}\right)$, and a polynomial-time variant achieving $O\left(\log n \cdot T^{\frac{2}{3}}(n\log T)^{\frac{1}{3}}\right)$.

## 2 Related Work

Traditional Multi-Armed Bandit (MAB) problems focus on maximizing a single cumulative reward Slivkins (2019), but many real-world applications involve multiple conflicting objectives. This complexity has led to the development of Multi-Objective Multi-Armed Bandits (MO-MAB), where the goal is to simultaneously maximize multiple objectives by pulling Pareto-optimal arms. A key challenge in this area is the evaluation of MO-MAB algorithms, particularly the design of an appropriate regret metric.

Drugan and Nowé 2013 introduced the first Pareto regret framework for MO-MAB, combining scalarization with an exploration-exploitation algorithm called *Pareto-UCB1*, which demonstrated logarithmic regret bounds. Their empirical studies Drugan et al. (2014) validated the algorithm on multi-objective Bernoulli distributions. However, challenges remain in balancing performance across objectives and ensuring diversity along the Pareto front. These limitations of the Pareto regret metric and Pareto-UCB1 are discussed in Section 3. Before delving into these issues, we provide an overview of related work and how recent studies have built on the Pareto regret framework.

Mahdavi et al. 2013 applied stochastic convex optimization techniques to MO-MAB, introducing scalarization functions (e.g., Chebyshev and linear scalarization) to address the complexity of optimizing multiple objectives under uncertainty. Yahyaa et al. 2014 developed the *Annealing-Pareto* algorithm, which adapts the annealing concept to MO-MAB, dynamically adjusting exploration intensity to improve the trade-off between exploration and exploitation. Yahyaa and Manderick 2015 applied Thompson sampling to MO-MAB, selecting arms based on their posterior distributions to manage uncertainty.

Busa-Fekete et al. 2017 proposed a MAB framework using the generalized Gini index, allowing the algorithm to prioritize better trade-offs and balance multiple objectives more effectively. They provided both theoretical analysis and empirical results showing improvements in exploration and exploitation. *Öner et al.* 2018 extended MAB by studying the combinatorial MO-MAB problem, where multiple arms can be selected at each round. Their algorithm combines exploration-exploitation strategies with combinatorial optimization to address conflicting objectives and constraints. Lu et al. 2019 developed a framework for generalized linear bandits, applying regret minimization techniques to handle multi-objective settings, offering both theoretical guarantees and empirical validation.

Xu and Klabjan 2023 extended the Pareto regret framework to adversarial settings in MAB, presenting algorithms for both stochastic and adversarial scenarios. However, their focus on optimizing a single objective led to suboptimal solutions in multi-objective contexts, where only one direction of the Pareto front was considered.

Turgay et al. 2018 integrated contextual information into MO-MAB, optimizing the decision-making process by considering contextual relationships between arms and objectives. Hüyük and Tekin 2021 developed an algorithm incorporating lexicographical ordering (prioritizing objectives) and satisfying (ensuring objectives exceed thresholds). Their analysis offered insights into improving the efficiency of MO-MAB. Xue et al. 2023 extended this work by generalizing lexicographically ordered MO-MAB from priority-based regret to general regret.

Cheng et al. 2024 proposed an algorithm that hierarchizes Pareto dominance to improve regret minimization by prioritizing objectives and refining exploration strategies. Ararat and Tekin 2023 introduced a framework based on a polyhedral ordering cone to define directional preferences among vector rewards, replacing traditional Pareto dominance, and established gap-dependent and worst-case sample complexity bounds for it. Crépon et al. 2024 studied the challenge of learning and identifying Pareto-optimal arms under the stochastic multi-armed bandit framework. They presented an algorithm to guarantee suboptimality relative to the true Pareto front. Building upon this, Karagözlü et al. 2024 addressed the challenge of learning the Pareto-optimal set under incomplete preferences in a pure exploration setting, providing an algorithm to identify Pareto-optimal arms.

In general, the approaches aiming to minimize Pareto regret, as defined by Drugan & Nowé (2013), either directly (e.g., Xu & Klabjan (2023)) or indirectly (e.g., Drugan et al. (2014)), often focus on optimizing along a single objective direction. While this can result in non-dominated solutions in one dimension, it fails to capture the entire Pareto front. To address these limitations, we propose a new regret metric that comprehensively evaluates performance across multiple objectives.

## 3 Analysis of Pareto Regret Used in the Literature

In 2013, Drugan et al. 2013 introduced the first regret metric for evaluating the efficiency of MO-MAB algorithms, known as *Pareto regret*. This metric quantifies, for each arm $a$, the distance between its reward vector $\mu(a)$ and the Pareto-optimal (PO) set $\mathcal{A}^*$. To do this, a *virtual reward vector $\nu(a)^*$* is constructed by adding a value $\epsilon$ to each objective of $a$ until it becomes *incomparable* (non-dominating) with any arm in $\mathcal{A}^*$. The regret $\Delta(a)$ is then defined as the difference between $\nu(a)^*$ and the original reward vector $\mu(a)$, effectively representing the minimum distance to the Pareto front. While this metric is simple and widely adopted, it has several notable limitations. The most significant issue is that it evaluates arms based solely on their distance to the Pareto front in one direction, without considering their performance across all objectives. This can lead to situations where algorithms optimizing a single objective achieve low regret even if they perform poorly on other objectives. For example, algorithms designed to prioritize one objective (e.g., see Xu & Klabjan (2023)) may appear effective under this metric despite failing to achieve a well-balanced multi-objective performance.

Figure 1 (left panel) illustrates this problem in a MO-MAB instance with three arms. Here, the Pareto UCB algorithm presented in Drugan & Nowé (2013) repeatedly (and fairly) selects the suboptimal arm with reward $(0,0)$, despite it being strictly dominated by arms $(1,0)$ and $(0,1)$. Indeed, the algorithm only does not select this dominated arm in the second round. Although the regret bound $\sum_{a \notin A^*} \frac{8 \cdot \log\left(T \sqrt[4]{D|A^*|}\right)}{\Delta_a}$, where $T$ is the number of plays and $D$ is the number of objectives, claimed by Drugan & Nowé (2013) holds, the choice of $\Delta_a$ as a small positive value can lead to poor practical performance. This issue is compounded in higher-dimensional settings, where the number of suboptimal arms grows exponentially with the number of objectives, dramatically reducing the probability of selecting even a PO arm in each round.

Additionally, Drugan et al. introduced a *fairness* concept to encourage more balanced performance across the PO arms. However, this metric assumes a uniform distribution of PO solutions along the Pareto front, a condition that is rarely met in real-world problems. For example, Figure 1 (right panel) shows a non-uniform Pareto front where fairness fails to ensure diversity. When the algorithm plays "fairly," as defined by Drugan & Nowé (2013), it may repeatedly select arms clustered near one extreme of the Pareto front, neglecting other promising regions.

These challenges highlight the shortcomings of the existing Pareto regret metric. Although it provides useful insights in certain cases, it does not adequately capture the trade-offs inherent in MO-MAB problems. Therefore, there is a clear need for a more comprehensive regret metric that evaluates algorithm performance holistically, accounting for both balance across objectives and diversity along the Pareto front.

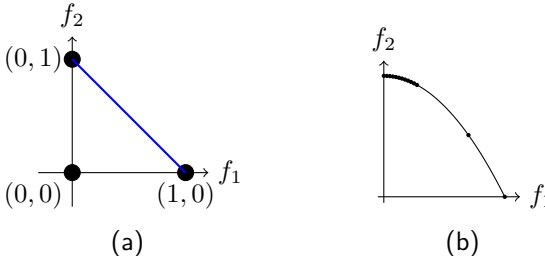

Figure 1: (a) MO-MAB instance with three arms: (1,0), (0,1), and (0,0). (b) PO solutions in a bi-objective maximization problem, not uniformly spread.

## 4 Multi-objective Multi-Armed Bandits

### 4.1 Domination and Efficient Pareto Optimal Arms

The main objective of solving multi-objective optimization problems is to identify a set of solutions that are close to the Pareto-Optimal (PO) solutions, while simultaneously ensuring a high degree of diversity within the objective space. This involves two orthogonal goals: first, to approximate the PO set as closely as possible, guaranteeing that the solutions reflect the best trade-offs among the competing objectives; and second, to maintain sufficient diversity among these solutions, covering various regions of the objective space. Such diversity is crucial for addressing different preferences among decision-makers effectively Coello (2007). These goals, and particularly the first one, can be extended naturally to online optimization and stochastic multi-objective multi-armed bandits (MO-MAB). Given an instance of a MO-MAB problem with $n$ arms, denoted by $\mathcal{A} = \{a_1, a_2, ..., a_n\}$, and $D$ (maximization) objectives $\mathcal{F} = \{f_1, f_2, \ldots, f_D\}$, let the reward vector of arm $a \in \mathcal{A}$ at time (or round) $t$ be denoted as:

$$\mathbf{r}^t(a) = (r_1^t(a), r_2^t(a), \ldots, r_D^t(a)),$$

where $r_d^t(a)$ is the reward of arm $a$ in the $d$-th objective (or dimension) at time $t$. In the stochastic setting

$$\sum_{t=1}^{T} \mathbf{r}^t(a) = (T\mu_1(a), T\mu_2(a), \ldots, T\mu_D(a)),$$

where $\mu_d(a)$ is the expected reward of arm $a$ in dimension $d$.

In the multi-objective context, an arm $a$ is said to *dominate* another arm $b$ (denoted as $a \succ b$) if and only if $a$ is at least as good as $b$ in all objectives, and is strictly better in at least one objective. Formally, for two reward vectors $\mathbf{r}^t(a) = (r_1^t(a), r_2^t(a), \ldots, r_D^t(a))$ and $\mathbf{r}^t(b) = (r_1^t(b), r_2^t(b), \ldots, r_D^t(b))$, arm $a$ dominates $b$ if:

- $r_d^t(a) \geq r_d^t(b)$   for all $d \in [D]$,   and

- $r_d^t(a) > r_d^t(b)$   for at least one $d \in [D]$,

where $[D]$ denotes the first $D$ positive integers, $[D] = \{1, 2, \ldots, D\}$. Similarly, $a$ is said to *weakly dominate* $b$ (denoted as $a \succeq b$) if and only if $a$ is at least as good as $b$ in all objectives, i.e., $r_d^t(a) \geq r_d^t(b)$   for all $d \in [D]$. So, an arm $a \in \mathcal{A}$ is a *non-dominated*, if there does not exist any other arm $b \in \mathcal{A}$ such that $b$ dominates $a$ (i.e., $\forall b \in \mathcal{A} : b \nsucc a$). Finally, the set of all non-dominated arms is called the *Pareto-optimal* (PO) set and denoted by $\mathcal{A}^*$. The image of PO arms in the objective space is called *Pareto Front*.

In online optimization and MAB, the iterative nature of the process shifts the objective from single-instance optimization to maximizing cumulative rewards over a given number of rounds. Unlike classical optimization, where a single decision is made, the decision-maker in this context selects (or pulls) from available choices iteratively over $T$ rounds. Therefore, we introduce the concept of the *Efficient Pareto-Optimal* (EPO) set,

which represents an efficient subset of the PO set, specifically tailored to maximize cumulative performance in an iterative decision-making setting. To this end, we need to extend the definition of domination to a subset of arms.

Let $S = \left(a^1, a^2, \ldots, a^T\right)$ be the sequence of arms selected by a decision-maker (arm $a^t$ at round $t \in [T]$ is pulled). So, the cumulative reward vector of $S$ is given by:

$$\bar{\mathbf{r}}(S) = \sum_{t=1}^{T} \mathbf{r}(a^t).$$

We can restrict the sequence of selected arms to any subset of arms. Let $X \subseteq \mathcal{A}^*$ and $Y \subseteq \mathcal{A}^*$ be two different subsets of PO arms, and $S_X$ and $S_Y$ be two sequences of arms selected (only) from $X$ and $Y$, respectively. We say $X$ weakly dominates $Y$, denoted $X \succeq Y$, if for any sequence $S_Y$, there exists a sequence $S_X$ such that the cumulative reward vector of $S_X$ weakly dominates that of $S_Y$. Formally,

$$X \succeq Y \iff \forall S_Y \in Y, \exists S_X \in X \text{ such that } \bar{\mathbf{r}}(S_X) \succeq \bar{\mathbf{r}}(S_Y).$$

Since the domination relation between sets involves convex combinations of arms with real-valued weights, while arm selections in each round are discrete (integer-valued), the domination relationship requires a sufficiently large number of rounds to achieve the desired weight approximation through integer pulls. More precisely, for the domination to hold, there must exist a threshold $T_0$ beyond which the integer constraints allow for a sufficiently accurate rational approximation of the required convex weights:

$$X \succeq Y \iff \forall S_Y \in Y, \exists S_X \in X\colon \qquad \exists T_0 \in \mathbb{N}, \forall T \geq T_0, \bar{\mathbf{r}}(S_X) \succeq \bar{\mathbf{r}}(S_Y),$$

where $T_0$ depends on the precision required to approximate the optimal convex weights through integer-valued arm pulls. In terms of the domination relation between sets of individual arms, two scenarios are possible: either one set dominates the other, or the sets are non-dominated with respect to each other. Now, we can define EPO set $\mathcal{E}\mathcal{A}^*$ as

$$\mathcal{E}\mathcal{A}^* = \{a^* \in \mathcal{A}^* | \nexists S \subseteq \mathcal{A}^* \setminus \{a^*\}, \bar{r}(S) \succeq \bar{r}(\{a^*\})\} \tag{1}$$

As an example, assume a simple instance of a MO-MAB problem with three PO arms $a$, $b$ and $c$ with reward vectors $\mathbf{r_a} = (1,0)$, $\mathbf{r_b} = (0,1)$, and $\mathbf{r_c} = (\epsilon, \epsilon)$ for some small positive value $\epsilon$. In this scenario, when a decision-maker pulls arms $a$ and $b$ iteratively, the average reward vector (the cumulative reward is a similar discussion) achieved is $\left(\frac{1}{2}, \frac{1}{2}\right)$. However, if the decision-maker pulls arm $c$, the resulting average reward vector would be $(\epsilon, \epsilon)$, which is lower than the first case for $\epsilon \leq \frac{1}{2}$. In this sense, $\{a, b\} \succ \{c\}$.

In this example, the positions of the arms form a symmetric configuration. For this symmetry reason, if arms $a$ and $b$ are pulled equally often, the cumulative reward vector $\{a, b\}$ *always* dominates the reward of arm $c$. In the general case, suppose arm $a$ is pulled $n_a$ times and arm $b$ is pulled $n_b$ times, where $n_a + n_b = T$. So, the resulting average reward vector is a linear combination of the reward vectors of $a$ and $b$ with the weights $w_a = \frac{n_a}{T}$ and $w_b = \frac{n_b}{T}$, respectively. As illustrated in Figure 2 (the left panel), this average vector will dominate any other arm whose reward vector is located within the gray region below and to the left of it. Although adjusting $n_a$ and $n_b$ can potentially cover the entire area below the line segment on the Pareto front between $a$ and $b$, for any fixed combination of $n_a$ and $n_b$, there remain two triangular regions that are not dominated by it. However, on the reverse side, for any arm $c$ below the Pareto segment connecting $a$ and $b$ on the Pareto front, there exists a specific range of values for $n_a$ and $n_b$ such that the average vector $\frac{n_a}{T}a + \frac{n_b}{T}b$ dominates $c$ (illustrated in the right panel of Figure 2).

Note that, in this example, $\{a, b\} \succ \{c\}$ holds while $c$ lies below the line segment. If $c$ lies above it, no pair of $n_a$ and $n_b$ will be able to dominate $c$. This means that no linear combination of $a$ and $b$ can dominate $c$.

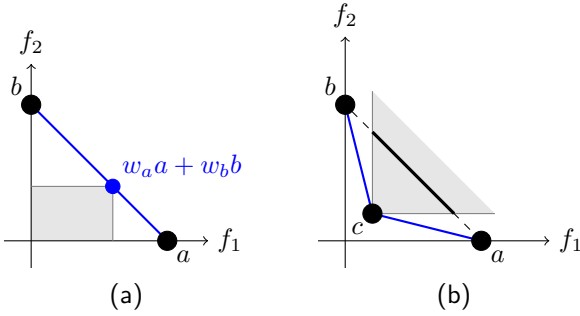

Figure 2: Dominance regions in MO-MAB. (a) Dominating region of a linear combination of two PO arms with rewards $(1,0)$ and $(0,1)$. (b) An MO-MAB instance with three arms: $a = (1,0)$, $b = (0,1)$, and $c = (0.2, 0.2)$. $c$ is dominated by linear combinations of $a$ and $b$ in the gray region.

In general, an arm $a$ at a given time $t$ is weakly dominated by some linear combinations of $a_1, a_2, \ldots, a_l$, if and only if, there exist $\alpha =< \alpha_1, \alpha_2, ..., \alpha_l >$ such that $\sum_{i=1}^{l} \alpha_i r^t(a_i) \succeq r^t(a)$, where $\sum_{i=1}^{l} \alpha_i = 1$.

Where, $r^t(a_i) = \mu(a_i)$ in the stochastic MO-MAB. Thus, the set $\mathcal{EA}^*$ represents those PO arms that lie in the convex position on the Pareto front. Therefore, after finding $\mathcal{A}^*$, the arms of $\mathcal{EA}^*$ can be efficiently found in polynomial time using a linear programming approach with $n^* - 1$ variables and $D$ constraints, where $n^* = |\mathcal{A}^*|$, and we do not need to find the convex hull of the arms which has exponential complexity. As a result of such a linear programming problem, if there is no feasible solution $\alpha = (\alpha_1, \alpha_2, ..., \alpha_l)$, arm $a$ belongs to $\mathcal{EA}^*$, otherwise, at least one vector $\alpha$ that $\sum_{i=1}^{l} \alpha_i r^t(a_i)$ weakly dominates $r^t(a)$ is computed.

**A practical example.** Multi-objective Vehicle Routing Problems (VRPs) constitute a major operational challenge for large-scale logistics providers such as DHL, Amazon, and national postal services, where 1,000–10,000 daily delivery tasks must be optimized under competing objectives with unknown priority weights. These include minimizing unassigned tasks, travel distance and duration, territorial overlap among routes, and soft-constraint violations (e.g., time-window penalties), while simultaneously maximizing route compactness—objectives that span incommensurable scales (monetary costs versus spatial quality concepts). At such scales, exact optimization is computationally prohibitive, rendering metaheuristic methods the only viable solution approach. Contemporary metaheuristics primarily employ ruin-and-recreate operators that partially dismantle incumbent solutions and reconstruct them using insertion heuristics. Modern systems combine diverse ruin strategies (e.g., cluster removal, worst-job removal, neighbor removal) with multiple recreate strategies (e.g., regret-based, cheapest, or gap-based insertion), producing more than sixty operator configurations whose performance is highly problem-dependent and shaped by spatial structure, temporal constraints, and regulatory requirements Vidal et al. (2012). Each configuration can be viewed as an arm in a multi-armed bandit framework, yielding multi-dimensional rewards across objectives. Unlike experimental testing phases, each iteration directly impacts operational performance—poor exploration translates to immediate costs and service degradation. The solver must therefore maintain diverse efficient strategies while minimizing cumulative regret over thousands of iterations.

Conventional Pareto-based multi-armed bandit formulations are insufficient in this context because they treat all non-dominated arms equivalently. Consequently, a configuration offering negligible (e.g., 0.01%) uniform improvements is regarded as equally valuable as one delivering substantial progress (e.g., a 0.5% reduction in distance) despite secondary objective trade-offs. Yet cumulative solution quality over generations depends on achieving meaningful aggregated gains. Two complementary Pareto-efficient strategies that each produce substantial improvements in distinct objectives can jointly yield far greater cumulative progress than uniformly weak Pareto strategies. The notion of efficient Pareto-optimal arms directly addresses this limitation by distinguishing genuinely promising configurations from those providing only trivial non-dominated gains.

## 4.2 Definition of Regret in MO-MAB

In MO-MAB problems, since there is more than one PO arm, the definition of *regret* is more complex. Most studies over the last decade have relied on the definition introduced in Drugan & Nowé (2013), which is discussed in Section 3. In this paper, we propose a more comprehensive definition of regret that is better suited to MO-MAB. Our definition is formulated to be applicable to both PO arms and EPO arms.

An algorithm $Alg$ that chooses an arm $a^t$ at time $t = 1, 2, \ldots, T$, has a regret $R$ compared to an arm $a^* \in \mathcal{A}^*$ (or alternatively an EPO arm $a^* \in \mathcal{E}\mathcal{A}^*$), if

$$\mathbb{E}\left[\sum_{t=1}^{T} r_d^t(a^*) - \sum_{t=1}^{T} r_d^t(a)\right] \le R, \quad \forall d \in [D].$$

In the stochastic setting, we can replace $\mathbb{E}\left[\sum_{t=1}^{T} r_d^t(a^*)\right]$ with $T\mu_d(a^*)$. Let denote this concept by $R_{Alg,a^*}(T) \le R$. Notably, the above definition of regret is stronger than the definition of "non-domination" used in Drugan & Nowé (2013). Here, $R_{\mathrm{Alg},a^*}(T)$ guarantees that the outcome is not only non-dominated compared to $a^*$ but also dominates it if it is improved by the regret value. Specifically, it is necessary to satisfy the inequality across all objectives, rather than just in one objective.

Now, let us extend $R_{\mathrm{Alg},a^*}(T)$ to the case where $Alg$ selects a set of arms $A^t$ at each round $t = 1, 2, \ldots, T$ and we compare it to all PO arms. In this case, we say that $Alg$ is $R$-regret, and denote it by $R_{\mathrm{Alg}}(T) \le R$, if the following regret properties are met:

## 1. Coverage-Regret

For any PO arm $a^* \in \mathcal{A}^*$ (or alternatively EPO arm $a^* \in \mathcal{E}\mathcal{A}^*$), there exists an arm $a^t \in A^t$ such that:

$$\mathbb{E}\left[\sum_{t=1}^{T} r_d^t(a^*) - \sum_{t=1}^{T} r_d^t(a^t)\right] \le R, \quad \forall d \in [D]. \tag{2}$$

or

$$T\mu_d(a^*) - \mathbb{E}\left[\sum_{t=1}^{T} r_d^t(a^t)\right] \le R, \quad \forall d \in [D]. \tag{3}$$

This property ensures that the algorithm can effectively approximate any PO (EPO) arm in terms of the reward vectors $r_d$ across all dimensions $d$ and over $T$ iterations.

## 2. Cumulative Adjustment-Regret

The cumulative minimal adjustment required for the arms in $A^t$ to weakly dominate some PO arm (or EPO arm) satisfies:

$$\mathbb{E}\left[\sum_{t=1}^{T} \sum_{a^t \in A^t} \min\{\epsilon \ge 0 \mid \exists a^* \in \mathcal{A}^* : a^t + \epsilon \succeq a^*\}\right] \le |\mathcal{A}^*|R.$$

Alternatively, we can replace $\mathcal{A}^*$ with $\mathcal{E}\mathcal{A}^*$.

This property bounds the cumulative regret associated with the adjustments needed for the arms selected by the algorithm to weakly dominate the optimal set $\mathcal{A}^*$ (or $\mathcal{E}\mathcal{A}^*$).

## Regret Interpretation

We define two complementary regret properties that jointly characterize algorithm performance: coverage-regret ensures completeness by requiring that all PO (or EPO) arms are well-approximated, while cumulative

adjustment-regret enforces minimality by penalizing unnecessarily large sets of selected arms. These two properties are complementary, as they measure orthogonal aspects of solution quality. The coverage-regret property evaluates how well the algorithm approximates the PO arms in the Pareto front. Specifically, it measures the discrepancy (in terms of rewards across all dimensions $d$) between a PO arm $a^* \in \mathcal{A}^*$ (or $a^* \in \mathcal{E}\mathcal{A}^*$) and the nearest arm $a^t \in A^t$. For each iteration $t$, the algorithm selects an arm $a^t \in A^t$ that minimizes:

$$\epsilon = \max_{d \in [D]} \left( r_d^t(a^*) - r_d^t(a^t) \right).$$

Thus, the selected arm $a^t$ ensures that the adjusted (virtual) arm $a^t + \epsilon$ weakly dominates $a^*$. The total expected regret over $T$ iterations is bounded by $R$, indicating the quality of the coverage provided by the algorithm for the Pareto front. As a complement, the cumulative adjustment-regret property evaluates the inverse relationship, quantifying the cumulative minimal adjustment required for the arms $A^t$ to weakly dominate the optimal set $\mathcal{A}^*$ (or $\mathcal{E}\mathcal{A}^*$). While a larger set $A^t$ reduces the coverage-regret, it increases the cumulative adjustment-regret due to the additional adjustments needed to align the selected arms with the optimal arms. This trade-off implies that $A^t$ should ideally approximate the Pareto front with a size proportional to $|\mathcal{A}^*|$, ensuring bounded cumulative adjustment-regret.

The interplay between coverage-regret and cumulative adjustment-regret highlights the balance required in selecting a representative set $A^t$. Increasing the size of $A^t$ improves coverage (reducing the coverage-regret) but may result in a higher cumulative adjustment-regret. To optimize performance, $A^t$ should approximate $\mathcal{A}^*$ as closely as possible while maintaining a manageable size, leading to a bounded cumulative adjustment-regret of $|\mathcal{A}^*|R$. For single-objective MAB problems, $D = 1$, the above definitions reduce to the classical notion of regret: both coverage-regret and cumulative adjustment-regret collapse to the standard formulation $\mathbb{E}\left[ \sum_{t=1}^{T} r^t(a^*) - \sum_{t=1}^{T} r^t(a^t) \right] \leq R$, where $a^*$ is the unique optimal arm. Our regret framework is comprehensive by design, requiring algorithms to compete against $\mathcal{A}^*$ across all dimensions $d = 1, \ldots, D$ simultaneously, rather than merely excelling in a subset of objectives. This stringent requirement distinguishes our approach from previous formulations, where superiority in just one dimension suffices for non-domination. While the bound appears dimension-independent in its functional form, the underlying algorithmic complexity scales with $D$ due to the requirement for comprehensive coverage. Furthermore, in multi-objective settings, the regret value in the coverage-regret property is non-negative when applied to $\mathcal{E}\mathcal{A}^*$ but may be negative for $\mathcal{A}^*$ due to the non-convexity of the Pareto front. This behavior arises because the iterative decision-making process in MAB allows selecting subsets of arms that collectively achieve superior rewards compared to individual non-convex PO arms.

## 5 A simple Multi-Objective MAB Algorithm

We introduce the first MO-MAB algorithm with a sublinear regret bound that satisfies both the coverage-regret and cumulative adjustment-regret properties, achieving a regret of $R = O\left( T^{\frac{2}{3}} (n \log T)^{\frac{1}{3}} \right)$. Additionally, we demonstrate that the algorithm's outcomes converge to the PO set as $T \to \infty$. The algorithm is initially explained and analyzed for PO arms, and we later extend the approach to efficiently handle EPO arms as well. This algorithm has two phases: In the exploration phase, each arm is explored by pulling it a fixed number of times ($T'$) to estimate the average reward for each arm. In the exploitation phase, the set of a minimum number of arms that cover all arms is identified ($B$) and pulled until iteration $T$. The pseudocode of the algorithm is represented below. The notation $a + 2r$ represents adding the scalar $2r$ to each dimension of arm $a$'s reward vector.

**Complexity.** After $T'$ rounds, Algorithm 1, in Step 5, removes all the arms that are clearly dominated by some other arms. This step takes $O(Dn^2)$ time and is valid because, at this stage, all arms are *clean* with a radius $r$, as established in Theorem 1. Importantly, no Pareto-optimal (PO) arms are discarded during this step. The primary purpose of Step 5 is to improve the algorithm's expected complexity. However, for worst-case analysis, this step can be omitted without affecting the theoretical regret guarantees of the algorithm.

---

**Algorithm 1** A Simple MO-MAB Algorithm

---

1: **Input:** Number of arms $n$, time horizon $T$
2: Set $T' = \left(\frac{T}{n}\right)^{\frac{2}{3}} (2 \log T)^{\frac{1}{3}}$
3: Pull each arm $T'$ times, and compute the average reward vector $\bar{\mu}(a)$ for all arms $a \in \mathcal{A}$
4: Set the confidence radius $r = \sqrt{\frac{2 \log T}{T'}}$
5: Remove all arms $a \in \mathcal{A}$ if there exists some $a' \in \mathcal{A}$ such that $a' \succeq a + 2r$
6: **for** any arm $a \in \mathcal{A}$ **do**
7:     Compute list $Dom(a) = \{a' \in \mathcal{A} : a + 2r \succeq a'\}$
8: **end for**
9: Compute the minimum set of arms $B$ such that $\bigcup_{b \in B} Dom(b) = \mathcal{A}$
10: **for** $t = T' + 1$ **to** $T$ **do**
11:     Pull all arms $b \in B$
12: **end for**
13: **return** $B$

---

Next, the algorithm computes $\text{Dom}(a)$ for each arm $a$, which represents the set of all arms $a'$ weakly dominated by $a + 2r$. Following this, the algorithm performs the minimum set cover computation of arms in Step 9. Specifically, it identifies the smallest number of improved arms, each enhanced by a factor of $2r$, that weakly dominate all other arms. This set is called the *minimum set covering* of arms.

The set cover problem is a well-known NP-hard problem Vazirani (2003), and in our scenario, it can be solved exactly with a time complexity of $O(2^n \cdot n)$ in the worst case, assuming no arms are removed in Step 5. Consequently, the overall complexity of Algorithm 1 is $O(2^n \cdot n)$. However, in the expected case, the number of PO arms is polylogarithmic in $n$ Bentley et al. (1978). This means the average running time of Algorithm 1 is $O(2^{(\log n)^{D-1}} \cdot (\log n)^{D-1})$ even for computing the exact minimum set cover, e.g., $O(n \log n)$ time for $D = 2$.

To address the worst-case time complexity of Algorithm 1, we can employ an approximation algorithm that achieves an approximation ratio of $O(\log n)$ Vazirani (2003). This allows us to reduce the complexity to polynomial time while obtaining a solution that is within a factor of $O(\log n)$ of the optimal set cover size. Therefore, we have two options:

1. Compute the optimal set cover in $O(2^n \cdot n)$ time.

2. Utilize an $O(\log n)-$approximation algorithm to obtain a set cover in polynomial time.

It is important to note that the approximate solution guarantees that the size of the set cover will be at most $O(\log n)$ times larger than the optimal solution. This trade-off between optimality and computational efficiency provides flexibility depending on the specific requirements and constraints of the application.

In the following, we demonstrate the impact of the two computational approaches on the complexity and regret of Algorithm 1. The first option, which computes the exact minimum set cover, results in an exponential-time algorithm with a regret bound of $R = O\left(T^{\frac{2}{3}} (n \log T)^{\frac{1}{3}}\right)$. The second option, employing the $O(\log n)-$approximation approach, yields a polynomial-time complexity algorithm with a regret bound of $R = O\left(\log n \cdot T^{\frac{2}{3}} (n \log T)^{\frac{1}{3}}\right)$.

The regret analysis of the proposed algorithm is summarized in the following theorems, with detailed proofs deferred to the Appendix.

**Theorem 1.** *The coverage-regret of algorithm 1 is $R = O\left(T^{\frac{2}{3}} (n \log T)^{\frac{1}{3}}\right)$.*

**Theorem 2.** *The outcome of Algorithm 1 converges to the PO arms $\mathcal{A}^*$ as $T \to \infty$.*

**Lemma 1.** *In case of the clean event $C$ happens, the optimal solution for the minimum set covering of arms, computed in Step 9 of Algorithm 1, is bounded by $|\mathcal{A}^*|$.*

**Theorem 3.** *The cumulative adjustment-regret is hold for algorithm 1 with the regret $R = O\left(T^{\frac{2}{3}}(n\log T)^{\frac{1}{3}}\right)$.*

**Theorem 4.** *In Algorithm 1, if after computing the minimum arm covering set $B$, the non-efficient arms are removed, Theorem 1, Theorem 2, and Theorem 3 remain valid for the efficient PO arms, $\mathcal{EA}^*$.*

Before concluding the theoretical results, we elucidate an additional property of the proposed MO-MAB algorithm, which pertains to the concept of *diversity* in classical multi-objective optimization problems (MOPs) Coello (2007), and its implications for computing regret in MAB approaches that compare the outcome with only one best arm. While our analysis focuses on all PO arms, the conclusions are equally applicable to EPO arms.

Diversity, a critical secondary objective in MOPs, concerns the challenge of identifying a representative subset of PO solutions that is as diverse as possible within the objective space. Since multi-objective optimization algorithms typically generate a limited subset of solutions from a potentially vast set of PO solutions, ensuring adequate diversity in this subset is crucial. The proposed Algorithm 1 addresses this by selecting the minimal set of covering arms, which are improved by a radius $r = \sqrt{\frac{2\log T}{T'}}$. This implies that if multiple PO arms exist within a specific region of the Pareto front, the algorithm prioritizes a minimum subset of these arms capable of dominating all others in the same region (when improved by a factor of $2r$).

Consider a (maximal) diverse set of PO arms, denoted as DPO, characterized by a radius $r = \sqrt{\frac{2\log T}{T'}}$. Define a virtual arm $b^*$ whose reward is the coordinate-wise average of the arms in DPO:

$$r_d(b^*) = \frac{1}{|\text{DPO}|} \sum_{a^* \in \text{DPO}} r_d(a^*), \quad \forall d \in [D].$$

The arm $b^*$ can be interpreted as the expected reward achievable by a decision-maker who restricts his choices to a diverse subset of Pareto-optimal arms within a specified radius $r$. This approach is an optimal strategy for efficiently covering all regions of the Pareto front. Another interpretation of this best arm $b^*$ is the cumulative reward if the decision-maker randomly chooses one Pareto-optimal arm in DPO in each round. Importantly, this definition extends the concept of selecting the "best" arm from single-objective multi-armed bandit (MAB) problems to multi-objective contexts. Unlike a simple weighted linear combination of objectives, this formulation focuses on the average reward across the diverse set of PO arms rather than directly aggregating the objective values.

Now, consider a version of Algorithm 1 that, after $T'$ rounds, and computing the minimum covering arms $B$, only one arm is randomly selected from $B$ and pulled for rounds $t = T'+1, T'+2, \ldots, T$. Thus, the number of all pulls will be $nT' + (T - T')$. In the following, we show that this version of the algorithm is an $R = O\left(T^{\frac{2}{3}}(n\log T)^{\frac{1}{3}}\right)$ compared to $b^*$ introduced above.

**Theorem 5.** *The regret of the single arm pulling version of Algorithm 1 compared to the average best arm $b^*$ is $R = O\left(T^{\frac{2}{3}}(n\log T)^{\frac{1}{3}}\right)$.*

*Proof.* The proof is similar to the poof of Theorem 3. Under the clean events, since all the arms are pulled in the first $T'$ rounds, then the regret value in the first $T'$ iteration of the algorithm is at most $nT'$ in total (as Term 1). Now, assume $b^t$ is the random arm selected from the computed minimum covering arms $B$, so,

$$\text{Term 2} = \mathbb{E}\left[\sum_{t=T'+1}^{T} \min\{\epsilon \geq 0 : b^t + \epsilon \succeq b^*\}\right].$$

As discussed, under the clean event condition, the arms in $B$ are the minimum covering DPO set. That means, improving each arm $b \in B$ with the radius $2r$ will dominate some Pareto arm in DPO. So, selecting uniformly random arms from $B$ in rounds $t = T'+1, T'+2, \ldots, T$ results in the expected sum of difference with $r_d(b^*)$ does not exceed $O(2r)$ for all $d \in [D]$. Therefore, the total regret can be written as

$$\mathbb{E}[T] \leq nT' + O((T - T')2r) \leq nT' + O(T2r),$$

where $c$ is a constant. Replacing $T' = \left(\frac{T}{n}\right)^{\frac{2}{3}} (2 \log T)^{\frac{1}{3}}$ and $r = \sqrt{\frac{2 \log T}{T'}}$ results in $\mathbb{E}[T] \leq O\left(T^{\frac{2}{3}} (n \log T)^{\frac{1}{3}}\right).$

$\square$

**Proposition.** When $T \to \infty$, so $r \to 0$, and DPD will equal all the Pareto-optimal arms. Thus, Theorem 5 holds for the definition of the *best* arm, which corresponds to the mean reward of the Pareto-optimal arms.

## 6 Experimental Results

While our primary contributions lie in the formal analysis of the proposed multi-objective bandit framework and its regret guarantees, we also provide an empirical assessment to illustrate the algorithm's practical behavior. All code was written in Python and executed on a laptop with an Intel Core i7-1165G7 CPU (2.80 GHz) and 16 GB of RAM.

For each problem size $(n, D, T)$, we generate a single set of "true" mean vectors $\{\mu_{a,d}\}_{a=1,\dots,n}^{d=1,\dots,D}$ by sampling uniformly from $[0, 1]$. During the rounds, we sample stochastic outcomes

$$R_{a,d}^t \sim \text{Bernoulli}(\mu_{a,d}),$$

for $a = 1, \dots, n$, $d = 1, \dots, D$, $t = 1, \dots, T$. At round $t = T'$, we run both the exact and greedy set-cover subroutines. This ensures that both methods operate on identical empirical data. As explained, the exact set cover algorithm exhaustively searches all subsets of candidate sets to be set $B$ in algorithm 1 as the smallest set of arms whose union covers all arms. It explores combinations of increasing size, ensuring optimality but requiring exponential time. The greedy approach iteratively selects the set that covers the largest number of uncovered arms until full coverage is achieved. It runs in polynomial time and guarantees optimality with a logarithmic approximation Vazirani (2003).

We conduct experiments for $n \in \{20, 50, 100\}$ and $D \in \{2, 3, 5\}$, and we choose the total horizon $T$ large enough so that the confidence radius is approximately 0.02. In theory, $r = \sqrt{2 \ln T / T'}$ ensures that adding $2r$ to each empirical mean yields valid dominance relations, but it can easily produce $r$ values near unity—collapsing the cover set, $B$, to size one. To achieve a practically meaningful margin (e.g. $r \approx 0.02$), $T'$ (and thus $T$) must grow by orders of magnitude. Consequently, one typically calibrates $r$ below its theoretical bound, accepting a slight relaxation of the guarantee in favor of a non-degenerate arm selection. For example, choosing $T = 10^8$ yields $T' \approx 10^5$ and $r \approx 0.02$. Each configuration is repeated independently 10 times to reduce sampling noise. We measure (i) the average true PO size, (ii) the average cover size $|B|$ in the algorithm, and (iii) the average wall-clock time for each case approximation and exact approach to compute the minimum set cover, reporting these values in Table 1.

Table 1 shows that as the number of arms $n$ or the number of objectives $D$ increases, the average Pareto-optimal set grows substantially (from roughly 4–5 arms at $D = 2$ to over 27 arms at $n = 100, D = 5$). Both exact and greedy set-cover methods consistently produce cover sets $|B|$ that are somewhat smaller than the true Pareto front, demonstrating effective reduction of candidate arms while preserving coverage. The greedy heuristic typically selects a cover of size within one arm of the exact solver (e.g., 7.5 vs. 7.2 at $n = 100, D = 3$), at a fraction of the runtime (orders of milliseconds versus seconds when $D$ and $n$ grow). These results confirm that the greedy approximation achieves near-optimal cover sizes in practice, with dramatic savings in computation time as problem dimensions scale.

Another notable feature of the algorithm is its economy in the exploitation phase: it identifies a small arm-set $B$ to pull after exploration. For instance, with $n = 100$, $D = 5$, and $T = 10^8$, the procedure dedicates only $T' = 97\,295$ rounds to exploration, then confines all remaining pulls to $|B| = 15$ arms—out of 27 true Pareto-optimal candidates. To illustrate this behavior for $D = 2$, Figure 3 presents a representative trial (for $n = 20, 50,$ and 100), plotting true means (black circles), PO arms (red stars), and the final cover set $B$ (blue circles). For example, in the right panel ($n = 100$), although 8 of the 100 arms are Pareto-optimal,

Table 1: Average Pareto-optimal set size, cover size $|B|$, and runtime for exact and greedy set-cover subroutines. All the instances are reported for $T = 10^8$ rounds.

| $n$ | $D$ | Avg. true PO | Exact set cover | | Greedy set cover | |
|---|---|---|---|---|---|---|
| | | | $|B|$ | Time (s) | $|B|$ | Time (s) |
| 20 | 2 | 4.3 | 3.2 | 0.0 | 3.2 | 0.0 |
| 20 | 3 | 7.2 | 4.8 | 0.1 | 4.9 | 0.0 |
| 20 | 5 | 14.2 | 12.6 | 0.6 | 12.7 | 0.0 |
| 50 | 2 | 4.8 | 2.6 | 0.0 | 2.6 | 0.0 |
| 50 | 3 | 10.5 | 6.0 | 0.2 | 6.1 | 0.0 |
| 50 | 5 | 18.2 | 11.2 | 6.1 | 11.5 | 0.1 |
| 100 | 2 | 5.6 | 2.2 | 0.0 | 2.2 | 0.0 |
| 100 | 3 | 14.1 | 7.2 | 4.95 | 7.5 | 0.0 |
| 100 | 5 | 26.7 | 15.1 | 29.0 | 15.7 | 0.2 |

the algorithm selects only 3 for pulling in the exploitation phase. By adding $2r$ to the empirical means of these three arms, the reduced cover set suffices to dominate all the other arms in the instance.

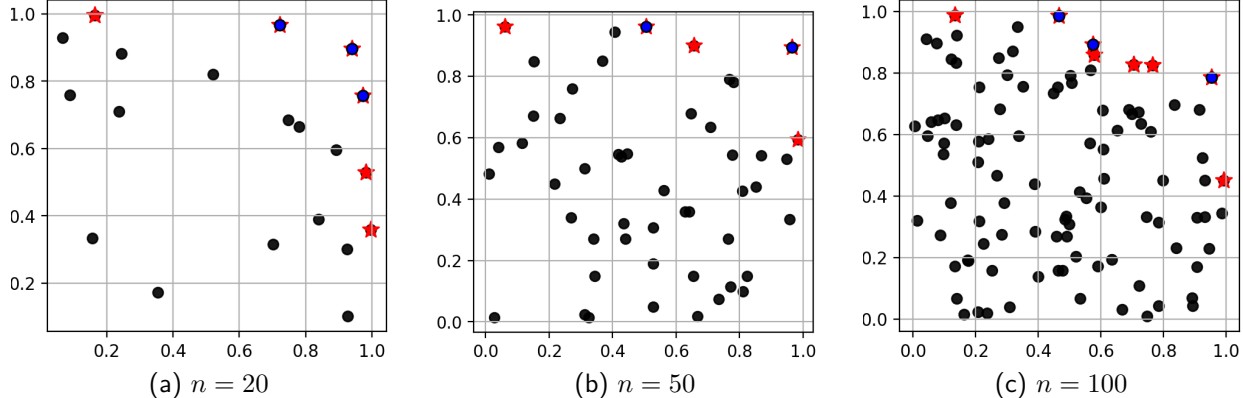

(a) $n = 20$        (b) $n = 50$        (c) $n = 100$

Figure 3: Empirical reward distributions and selected cover sets for three problem sizes with $D = 2$. The axes represent the reward values in each dimension. Black circles indicate the true mean rewards of arms, red stars denote the PO arms, and blue circles highlight the arms selected by the algorithm to form set $B$.

## 7 Conclusion

This paper introduced a novel regret metric for multi-objective multi-armed bandits (MO-MAB) designed to overcome the limitations of existing metrics by comprehensively accounting for all objectives simultaneously. We further defined the concept of efficient Pareto-optimal arms as those residing on the convex hull of the Pareto-optimal front. Leveraging this metric, we proposed a new algorithm proven to achieve sublinear regret for both Pareto-optimal and efficient Pareto-optimal arms in stochastic environments.

While this work presents the first algorithm rigorously evaluated under the proposed regret framework, our analysis reveals limitations in its theoretical regret bound and computational complexity. Future research should prioritize developing algorithms that achieve tighter regret guarantees, potentially through alternative algorithmic designs or refined analytical techniques. Moreover, the regret framework established here provides a formal foundation for extending the analysis to the significantly more challenging setting of adversarial MO-MAB, making the development of efficient algorithms robust to non-stochastic rewards a critical and promising direction for advancement in multi-objective online decision-making.

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

# A    Appendix

**Theorem 1.** *The coverage-regret of algorithm 1 is $R = O\left(T^{\frac{2}{3}}(n \log T)^{\frac{1}{3}}\right)$.*

*Proof.* Let $a^*$ be an arm in $\mathcal{A}^*$. The goal is constructing a sequence of arms $S = \{a^t\}_{t=1}^T$ where $a^t$ is pulled by the algorithms in rounds $t = 1, 2, \ldots, T$ such that

$$\mathbb{E}\left[\sum_{t=1}^T r_d^t(a^*) - \sum_{t=1}^T r_d^t(a^t)\right] \leq O\left(T^{\frac{2}{3}}(n \log T)^{\frac{1}{3}}\right), \forall d \in [D].$$

To construct $S$, we simply choose arm $a^*$ in the first $T'$ rounds, and for the next rounds ($t = T'+1$ to $t = T$) we choose $a^*$ if it is not removed from $B$. Otherwise, we choose an arm $b \in B$ such that $b + 2r \succeq a^*$.

For such a sequence of arms, if $a^* \in B$, the regret will be zero. Thus, let's analyze the case of $a^* \notin B$, that is, there exists an arm $b \in B$ such that $b + 2r \succeq a^*$.

We define the clean event Slivkins (2019) as

$$C = \left\{\left|\bar{\mu}_d^t(a) - \mu_d(a)\right| \leq r, \forall a \in \mathcal{A}, \forall d \in [D], \forall t \in [T]\right\},$$

where the confidence radius $r = \sqrt{\frac{2 \log T}{T'}}$. Define the *unclean event* $\overline{C}$ as the complement of the clean event $C$. By the law of total expectation,

$$\mathbb{E}[R(T)] = \mathbb{E}[R(T)|C]\,P[C] + \mathbb{E}[R(T)|\overline{C}]\,P[\overline{C}].$$

Applying Hoeffding's inequality and union bounds, the deviation of the average reward from the true expected reward can be bound as:

$$P[\overline{C}] \leq \frac{2nDT}{T^4} \leq \frac{2}{T}.$$

We used the fact that $n < T$ and $D < T$. So, the term $\mathbb{E}[R(T)|\overline{C}]\,P[\overline{C}]$ is negligible. Thus, by the fact that $P[C] \leq 1$, we can focus only on the clean events, particularly, $\mathbb{E}[R(T)|C]$.

Since arm $b + 2r$ dominates arm $a^*$, and both arms $a$ and $b$ are clean, we can write

$$\bar{\mu}_d(a^*) \leq \bar{\mu}_d(b) + 2r, \forall d \in [D],$$

and

$$\mu(a^*) \leq \bar{\mu}(a^*) + r, \text{ and } \mu(b) \leq \bar{\mu}(b) + r.$$

Thus,

$$\mu(a^*) - \mu(b) \leq 4r = 4\sqrt{\frac{2\log T}{T'}}.$$

Considering all $T - T'$ rounds, results in

$$E[R(T)|C] \leq (T - T')4\sqrt{\frac{2\log T}{T'}} < 4T\sqrt{\frac{2\log T}{T'}}.$$

By choosing $T' = \left(\frac{T}{n}\right)^{\frac{2}{3}} (2\log T)^{\frac{1}{3}}$, we have

$$E[R(T)|C] \leq 4T^{\frac{2}{3}}(2n\log T)^{\frac{1}{3}}.$$

Finally,

$$\begin{aligned}
\mathbb{E}[R(T)] &= \mathbb{E}[R(T)|C]\, P[C] + \mathbb{E}[R(T)|\overline{C}]\, P[\overline{C}] \\
&\leq 4T^{\frac{2}{3}}(2n\log T)^{\frac{1}{3}} + (T - T')\frac{2}{T} \\
&= O\left(T^{\frac{2}{3}}(n\log T)^{\frac{1}{3}}\right)
\end{aligned}$$

$\square$

**Theorem 2.** *The outcome of Algorithm 1 converges to the PO arms $\mathcal{A}^*$ as $T \to \infty$.*

*Proof.* The outcome of Algorithm 1 is the set $B$ which remains unchanged after round $T'$. To prove this convergence property, we need to show that as the time horizon $T$ approaches infinity, the probability of $B = \mathcal{A}^*$ approaches 1. When $T$ approaches infinity, $T' = \left(\frac{T}{n}\right)^{\frac{2}{3}} (2\log T)^{\frac{1}{3}}$ approaches infinity and consequently the confidence radius $r = \sqrt{\frac{2\log T}{T'}}$ approaches zero. Therefore, none of the PO solutions is dominated by some other arms $b + 2r \approx b$ in the second phase of algorithm 1. So, all the PO arms will appear in $B$. On the other hand, by approaching the confidence radius $r$ to zero, any non-PO arm at least must be dominated by some PO arm. As a result, set $B$ will be exactly all PO arms.

$\square$

**Lemma 1.** *In case of the clean event $C$ happens, the optimal solution for the minimum set covering of arms, computed in Step 9 of Algorithm 1, is bounded by $|\mathcal{A}^*|$.*

*Proof.* Note that no PO arms are removed up to Step 9 in the algorithm. On the other hand, the union of PO arms dominates all other arms. When all arms are clean with radius $r$, the union of improved (by $2r$) PO arms, also dominates all other arms, i.e., $\bigcup_{a^* \in \mathcal{A}^*} Dom(a^*) = \mathcal{A}$.

Therefore, the minimum set covering of arms has at least one solution of size $|\mathcal{A}^*|$. Consequently, the optimal solution is bounded by $|\mathcal{A}^*|$.
$\square$

Now let's prove the cumulative adjustment-regret bound for algorithm 1.

**Theorem 3.** *The cumulative adjustment-regret holds for algorithm 1 with the regret $R = O\left(T^{\frac{2}{3}}(n\log T)^{\frac{1}{3}}\right)$.*

*Proof.* To prove this theorem, we need to show that

$$\mathbb{E}\left[\sum_{t=1}^{T}\sum_{a^t\in A^t}\min\left\{\epsilon\geq 0\mid \exists a^*\in\mathcal{A}^*: a^t+\epsilon\succeq a^*\right\}\right]$$
$$\leq |\mathcal{A}^*|O\left(T^{\frac{2}{3}}\left(n\log T\right)^{\frac{1}{3}}\right).$$

Similar to Theorem 1, the probability of the unclean event $P[\overline{C}]$ is bounded by $\leq \frac{2}{T}$. Therefore, we assume only the case where all arms are clean. For this case, we decompose the above right-hand term into two components to analyze them separately and derive the desired bounds.

$$\text{Term 1} = \mathbb{E}\left[\sum_{t=1}^{T'}\sum_{a^t\in A^t}\min\{\epsilon\geq 0\mid \exists a^*\in\mathcal{A}^*: a^t+\epsilon\succeq a^*\}\right]$$

and

$$\text{Term 2} = \mathbb{E}\left[\sum_{t=T'+1}^{T}\sum_{a^t\in A^t}\min\{\epsilon\geq 0\mid \exists a^*\in\mathcal{A}^*: a^t+\epsilon\succeq a^*\}\right]$$

Based on the behavior of Algorithm 1,

$$\text{Term 1} = \sum_{t=1}^{T'}\sum_{a\in\mathcal{A}}\min\{\epsilon\geq 0\mid \exists a^*\in\mathcal{A}^*: a^t+\epsilon\succeq a^*\}$$

That is pulling all arms in $\mathcal{A}$. So,

$$\text{Term 1} \leq \sum_{t=1}^{T'}\sum_{a\in\mathcal{A}}1 \leq nT' = O\left(T^{\frac{2}{3}}(n\log T)^{\frac{1}{3}}\right).$$

We now proceed to bound

$$\text{Term 2} = \mathbb{E}\left[\sum_{t=T'+1}^{T}\sum_{b\in B}\min\{\epsilon\geq 0\mid \exists a^*\in\mathcal{A}^*: b+\epsilon\succeq a^*\}\right],$$

where $B$ is the minimum set of arms such that every arm $a\in\mathcal{A}$, including PO arms, is weakly dominated by $b+2r$ for some $b\in B$. If $b$ is a PO arm, then $\epsilon=0$. Otherwise, we show that under these conditions, there always exists a PO arm $a^*$ such that $a^*\succeq b$ and $\mu_d(a^*)-\mu_d(b)\leq 4r$ for all $d\in[D]$. Specifically, this $a^*$ can be expressed as:

$$a^* = \underset{a\in\mathcal{A}^*\text{ and }a\succeq b}{\operatorname{argmin}}\ \max_{k\in[D]}\mu_d(a)-\mu_d(b).$$

Assume, for contradiction, that there exists some $k\in[D]$ such that $\mu_d(a^*)-\mu_d(b)>4r$. Since both $b$ and $a^*$ are clean ($\forall d\in[D]: |\mu_d(a^*)-\bar{\mu}_d(a^*)|\leq r$ and $|\mu_d(b)-\bar{\mu}_d(b)|\leq r$), this implies $b\in Dom(a^*)$ but $a^*\notin Dom(b)$. However, as the weak domination relation ($\succeq$) is transitive, it follows that $Dom(b)\subset Dom(a^*)$.

Given that $B$ is the minimum set covering all arms:

- If $a^*\in B$, then $b\notin B$, as including both would violate $B$ is the minimum set cover.

- If $a^*\notin B$, there must exist some clean arm $b'\in B$ such that $a^*\in Dom(b')$. By transitivity, this implies $b\in Dom(b')$, again contradicting $B$ is the minimum set cover.

Thus, both cases lead to contradictions, proving that $\mu_d(a^*) - \mu_d(b) \leq 4r$ for all $d \in [D]$.

So, we can bound Term 2 as:

$$\text{Term } 2 \leq \mathbb{E}\left[\sum_{t=T'+1}^{T}\sum_{b \in B} 4r\right] = (T - T')|B|4\sqrt{\frac{2\log T}{T'}}.$$

This simplifies further to:

$$\text{Term } 2 < 4T|B|\sqrt{\frac{2\log T}{T'}}.$$

Based on lemma 1, $|B| \leq |\mathcal{A}^*|$. So, by replacing $T' = \left(\frac{T}{n}\right)^{\frac{2}{3}}(2\log T)^{\frac{1}{3}}$, we obtain:

$$\text{Term } 2 \leq O\left(|\mathcal{A}^*|T^{\frac{2}{3}}(n\log T)^{\frac{1}{3}}\right).$$

Finally,

$$\text{Term } 1 + \text{Term } 2 \leq |\mathcal{A}^*|O\left(T^{\frac{2}{3}}(n\log T)^{\frac{1}{3}}\right).$$

While the main proof concludes here, we note that the polynomial-time approximation for the minimum covering set satisfies $|B| \leq \log n \cdot |\mathcal{A}^*|$. Consequently, the polynomial-time algorithm achieves the regret bound of $O\left(\log n \cdot T^{\frac{2}{3}}(n\log T)^{\frac{1}{3}}\right)$ while maintaining correctness.

$\square$

**Theorem 4.** *In Algorithm 1, if after computing the minimum arm covering set B, the non-efficient arms are removed, Theorem 1, Theorem 2, and Theorem 3 remain valid for the efficient PO arms, $\mathcal{E}\mathcal{A}^*$.*

*Proof.* An arm $a \in B$ is identified and removed as a non-efficient arm if a linear combination of other arms in $B$ weakly dominates $a$. Specifically, in this case, there exists a subset $B' \subseteq B$ with $m \leq n$ arms and coefficients $\alpha = (\alpha_1, \alpha_2, \ldots, \alpha_m)$, such that $\sum_{i=1}^{m}\alpha_i = 1$, where the *artificial arm* $b_\alpha = \sum_{i=1}^{m}\alpha_i a_i$ dominates $a$. The term "artificial" reflects that no individual arm has rewards identical to $b_\alpha$, but since Theorem 1, and Theorem 3 discuss the expected value for the defined regret, the rewards of $b_\alpha$ can be approximated for sufficiently large $T$ by selecting arm $a_i \in B'$ with probability $\alpha_i$. Replacing $b$ with such an artificial arm $b_\alpha$ in Theorems 1 and 3 confirms their validity for the efficient PO arms, $\mathcal{E}\mathcal{A}^*$. For Theorem 2, since $r = \sqrt{\frac{2\log T}{T'}}$ converges to zero as $T \to \infty$, removing non-efficient arms from $B$ ensures only efficient PO arms remain in $\mathcal{E}\mathcal{A}^*$. $\square$

