# Stochastic Multi-Objective Multi-Armed Bandits: Regret Definition and Algorithm
## paper ID: 6238
## Response to reviewer #3

## Response to Reviewer

We sincerely thank the reviewer for their thoughtful and constructive feedback. We greatly appreciate the recognition of our framework's innovation and the recommendation for acceptance upon addressing the comments. We have carefully revised the manuscript to address all the requested changes, and we believe these revisions have strengthened the paper.

## Summary of Revisions

**Comment 1:** The authors claim that the existing works do not consider optimizing in the Pareto front but often focusing on optimizing along a single objective direction. The authors give two main properties of their proposed regret definition: coverage-regret property and cumulative adjustment-regret property. Could the authors please clarify how these two properties are necessary for a regret definition to be meaningful in the multi-objective MAB setting? For example, what if a regret definition only satisfies one of these properties but not the other? Would it still be meaningful?

**Answer:** The two properties are complementary and both necessary for meaningful multi-objective regret evaluation. Coverage-regret alone would allow algorithms to achieve low regret by selecting excessively large arm sets that trivially cover all Pareto-optimal arms, leading to inefficient exploration. Conversely, cumulative adjustment-regret alone would permit algorithms to select minimal arm sets that may leave significant gaps in Pareto front coverage. For example, an algorithm satisfying only coverage-regret might pull all $n$ arms equally, achieving perfect coverage but poor efficiency. An algorithm satisfying only cumulative adjustment-regret might focus on a single Pareto-optimal arm, achieving minimal adjustment cost but failing to capture the diversity essential for multi-objective optimization. The combination ensures both comprehensive coverage and efficient arm selection. **We addressed this clarification in subsection "Regret Interpretation" in Section 4 of the revision.**

**Comment 2:** In the proof of Theorem 1, $P[\overline{C}]$ should be bounded by $\frac{2nDT}{T'^4}$ instead of $\frac{2nDT}{T^4}$ based on the choice of is $r = \sqrt{\frac{2\log T}{T'}}$. Could the authors please elaborate on this?

**Answer:** In fact, the bound is correct, just some intermediate steps in the derivation of Hoeffding's inequality were omitted for brevity. We provide the detailed derivation below.

Applying Hoeffding's inequality with union bounds over all arms, dimensions, and time steps, we obtain:

$$P[\overline{C}] \leq 2nDT \cdot e^{-2T'r^2}$$

Substituting our choice of confidence radius $r = \sqrt{\frac{2\log T}{T'}}$:

$$P[\overline{C}] \leq 2nDT \cdot e^{-2T'\left(\sqrt{\frac{2\log T}{T'}}\right)^2} = 2nDT \cdot e^{-2T' \cdot \frac{2\log T}{T'}}$$

Simplifying the exponent, the $T'$ terms cancel:

$$P[\overline{C}] \leq 2nDT \cdot e^{-4\log T} = 2nDT \cdot e^{\log T^{-4}} = \frac{2nDT}{T^4}.$$

The key insight is that while the initial bound appears to scale as $2nDT$, the exponential decay with $T^4$ in the denominator dominates, yielding the tight bound $O(1/T)$ that makes the unclean event probability negligible.

This confirms that our analysis correctly bounds the probability of deviation from the clean event, ensuring the regret analysis holds with high probability.

**Comment 3:** Some typos to fix throughout the paper:

- In key contributions, the third bullet point is repeated twice. Please fix this.

- In related work, many citations are not in the correct format. For example, Yahyaa et al. Yahyaa et al. (2014) $--$ > Yahyaa et al., 2014. Yahyaa and Manderick Yahyaa & Manderick (2015) $--$ > Yahyaa and Manderick, 2015. Please fix this throughout the paper.

- The grammar in many sentences needs to be improved. For example, in the statement of Theorem 1, "The cumulative adjustment-regret is hold" could be "The cumulative adjustment-regret holds". At the beginning of section 5, "The following introduces the first MO-MAB algorithm with a sublinear regret bound" could be "We introduce the first MO-MAB algorithm with a sublinear regret bound". Please fix this throughout the paper.

**Answer:** We thank for these detailed editorial suggestions. We have addressed all mentioned issues in the revision: removed the duplicate bullet point in the Introduction, corrected all citation formatting inconsistencies, and fixed grammatical errors. Additionally, we conducted a thorough review of the manuscript to enhance overall writing quality and consistency.