# OpenReview forum: "Stochastic Multi-Objective Multi-Armed Bandits: Regret Definition and Algorithm"
_TMLR — Accepted by TMLR_

### Review · Reviewer_7NwK · 2025-12-06

**Summary Of Contributions:**

This paper introduces a new framework for the problem of multi-objective multi-armed bandits, in which the agent is allowed to pull multiple arms at each round. The regret formulation goes beyond the traditional Pareto-optimality criterion and is instead defined with respect to a newly proposed notion of Efficient Pareto Optimality. Building on this framework, the authors further develop an algorithm that achieves sublinear regret.

Key Strength:

1. The proposed framework is highly innovative.

Key Weaknesses:

1. The proposed framework lacks a (relatively) clear practical motivation.

2. The core idea of the algorithm largely follows the standard confidence-bound approach used in methods such as L/UCB. Although the authors show that the proposed algorithm achieves sublinear regret, the paper does not provide a tightness analysis (e.g., a corresponding lower bound). That said, given that this is the first work to introduce this new framework, this limitation is understandable.

**Additional Comments:**

Indeed, restricting attention solely to Pareto optimality is highly limiting. The research direction introduced by the authors—namely efficient Pareto optimality—is inherently meaningful. In particular, the illustrative example with three arms (i.e., (1,0),(0,1), (\epsilon,\epsilon)) is quite insightful.

However, given that the technical contribution of the proposed algorithm is rather limited (as discussed in the “Summary of Contributions”), the primary value of this work lies in the introduction of the new framework itself. For this reason, it is essential to provide a practical example (even a relatively simple or stylized one) to better motivate and demonstrate the utility of the proposed framework.

That said, I would recommend acceptance of this paper once the practical example is clear.

**Audience:**

Yes

**Audience Explanation:**

The proposed new framework of MO-MAB is quite promising.

**Claims And Evidence:**

Yes

**Claims Explanation:**

I did not examine the proofs in full detail, but the techniques appear to follow the standard L/UCB methodology, and the overall seems sound.

**Requested Changes:**

1. On page 5, the phrase “for a sufficiently large number of rounds” is overly vague. This needs to be clarified. Since the primary contribution of the paper lies in the formulation of the new problem, the problem statement itself must be precisely written so that future researchers can accurately follow and build upon it.

2. It is unclear why both Cumulative Adjustment-Regret and Coverage-Regret are discussed together in Section 4. This presentation makes it somewhat difficult to identify the main focus. Note that different regret notions in bandit problems often reflect conflicting objectives. For example, in the classical bandit setting, an algorithm that is optimal in terms of simple regret may not be optimal with respect to cumulative regret, and vice versa.

Therefore, I suggest the following:
(a) If the authors can show that the two regret notions are fundamentally equivalent—for instance, by proving that an algorithm optimal for one type of regret (or an adaptation of it) is also optimal for the another type of regret —then keeping both regret formulations in the same section is justified.

(b) Otherwise, the main text should focus on a single regret notion, while the discussion of the other should be moved to the appendix as "more discussion."

3. It would be better to provide a practical example for the proposed framework.

---

### Review · Reviewer_fWd1 · 2025-12-17

**Summary Of Contributions:**

The paper makes several distinct contributions to the field of Multi-Objective MAB:
1. The authors propose a new, comprehensive regret framework consisting of "Coverage-Regret" and "Cumulative Adjustment-Regret". This framework aims to address the limitations of the existing Pareto regret metric.
2. The paper introduces the concept of EPO arms, which are Pareto-optimal arms lying on the convex hull of the Pareto front. This concept is tailored for online optimization, leveraging the iterative nature of bandit problems to argue that mixtures of arms can dominate non-convex Pareto solutions.
3. Algorithm design and analysis. The authors develop a two-phase "explore-then-exploit" algorithm with a theoretical regret guarantee of $O(T^{2/3}(n\log T)^{1/3})$. It also proves that the algorithm's outcome converges to the Pareto-optimal set as the time horizon approaches infinity.

Stengths:
1. The MO-MAB problem is interesting and practically relevant.
2. The proposed metrics and methods are useful to some extend and overcome some challenges of the previous literature.
3. The paper is relatively easy to follow.

Weaknesses:
1. The theoretical results are somewhat weak. The $T^{2/3}$-regret should not be optimal. An educated guess of the tight regret should be $\sqrt{T}$ as the  $T^{2/3}$ regret does not depend on $D$. If this intuition is wrong, the regret dependence on $D$ is very interesting to see and understand.
2. The numerical results do not include some informative comparative baselines. We do not see any results from the state-of-art MO-MAB algorithms.
3. The motivating example in the revised version is fine but not perfect. The scenario described sounds more like a best arm identification problem instead of regret minimization problem. In practice, people usually have multiple configurations to test with the purpose to choose the one that is the best, and thus the regret in the testing periods may not be very important.

**Audience:**

Yes

**Audience Explanation:**

MAB is one of the core topics in ML, and this paper lies naturally in the scope of TMLR. Besides, the paper is relevant to the part of TMLR's audience working on the theoretical foundations of multi-objective learning.

**Broader Impact Concerns:**

I think the paper is fine with this part.

**Claims And Evidence:**

Yes

**Claims Explanation:**

I think the paper is well organized and the results are clearly presented and supported, although there is some space to improve.

**Requested Changes:**

Following my comments about weakness above, there are several changes from my perspective which can make the paper better.

1. The dependence of the regret results on dimension $D$. Particularly, I am curious about the case $D=1$ and its connection to the conventional MAB problem.
2. Some comparison with some baseline algorithms in the literature can be helpful since the authors point out some of the disadvantages of the formulation of the paper in the literature.
3. Polish the motivating example slightly better. The current version should be fine as well, but can be better. This is a very minor point.

---

### Review · Reviewer_hg41 · 2025-12-20

**Summary Of Contributions:**

The paper studies stochastic multi-objective multi-armed bandits (MO-MAB), where each arm yields a vector-valued reward across multiple conflicting objectives. The learner must balance exploration and exploitation while accounting for trade-offs among objectives.
Most prior MO-MAB work evaluates algorithms using the Pareto regret metric introduced by Drugan \& Nowé (2013). The authors argue that this metric is fundamentally flawed because it measures distance to the Pareto front only in a single direction, and it results in an algorithm that achieves low regret that optimizes only one objective and fails to ensure diversity across the Pareto front.
Therefore, the authors propose two new regrets: (1) coverage-regret, ensuring all Pareto-optimal arms are well approximated, and (2) cumulative adjustment-regret, penalizing selecting unnecessarily large arm sets.
The authors give an arm elimination-based algorithm and measure the regret in using these two new metrics and prove that the algorithm achieves regret $O(T^{2/3} (n \log(T))^{1/3})$ in both metrics, where $n$ is the number of arms and $T$ is the time horizon.
The authors also provide empirical validation of their algorithm on synthetic experiments with Bernoulli rewards, showing that the greedy approximation yields cover sizes close to optimal, runtime scales favorably compared to exact set cover, and the exploitation phase focuses on a small subset of arms while preserving Pareto coverage.

**Audience:**

Yes

**Audience Explanation:**

Yes, this paper proposes an interesting topic in multi-objective multi-armed bandits. The proposed regret definitions and algorithm could be of interest to researchers working on multi-objective optimization and bandit problems.

**Broader Impact Concerns:**

The submission is theoretical in nature, focusing on regret definitions, algorithmic design, and analysis for stochastic multi-objective multi-armed bandits. This paper does not introduce immediate societal, ethical, or safety risks.

**Claims And Evidence:**

Yes

**Claims Explanation:**

The authors give a clear statement of the problem setting. However, the motivation for the two proposed regret definitions could be further clarified. For example, why are the coverage-regret property and cumulative adjustment-regret property necessary for a regret definition to be meaningful in the multi-objective MAB setting? The authors could provide more intuition on why these two properties are important.
Additionally, the proof of Theorem 1 contains a minor error in bounding $P[\bar{C}]$. The authors should clarify this in the revision.

**Requested Changes:**

The authors claim that the existing works do not consider optimizing in the Pareto front but often focusing on optimizing along a single objective direction.

The authors give two main properties of their proposed regret definition: coverage-regret property and cumulative adjustment-regret property. Could the authors please clarify how these two properties are necessary for a regret definition to be meaningful in the multi-objective MAB setting? For example, what if a regret definition only satisfies one of these properties but not the other? Would it still be meaningful?

In the proof of Theorem 1, $P[\bar{C}]$ should be bounded by $\frac{2n D T}{T'^4}$ instead of $\frac{2n D T}{T^4}$ based on the choice of $r$ is $\sqrt{\frac{2 \log(T)}{T'}}$. Could the authors please elaborate on this?

Some typos to fix throughout the paper:

1. In key contributions, the third bullet point is repeated twice. Please fix this.

2. In related work, many citations are not in the correct format. For example, Yahyaa et al. Yahyaa et al. (2014) -> Yahyaa et al., 2014. Yahyaa and Manderick Yahyaa & Manderick (2015) -> Yahyaa and Manderick, 2015.
Please fix this throughout the paper.

3. The grammar in many sentences needs to be improved. For example, in the statement of Theorem 1, "The cumulative adjustment-regret is hold" could be "The cumulative adjustment-regret holds". At the beginning of section 5, "The following introduces the first MO-MAB algorithm with a sublinear regret bound" could be "We introduce the first MO-MAB algorithm with a sublinear regret bound".
Please fix this throughout the paper.

---

### Decision · Action_Editor_HRrm · 2026-02-02

**Recommendation:** Accept as is

**Additional Comments:**

The paper proposes a novel framework for measuring regret in Multi-Objective Multi-Armed Bandit settings. Existing metrics popularly used for computing regret in this setting tend to focus on Paretto efficiency in the direction of one particular objective, instead of more broadly considering all objectives.The reviewers all agree the framework is novel and interesting and the evidence adequately supports the claim and there is clearly an audience for this work in the community, this paper is recommended for acceptance.

**Audience:**

Yes

**Audience Explanation:**

All reviewers consider the framework novel and interesting.

**Claims And Evidence:**

Yes

**Claims Explanation:**

The paper proposes a novel framework for measuring regret in the Multi-Objective MAB problem. They present the shortcomings of the existing approach for evaluating regret in this setting and provide an algorithm optimizing for their proposed metric, accompanied by theoretical guarantees and an experiment on artificial data.